# Evaluating the RIST Molecular-Targeted Regimen in a Three-Dimensional Neuroblastoma Spheroid Cell Culture Model

**DOI:** 10.3390/cancers15061749

**Published:** 2023-03-14

**Authors:** Carina Kaess, Marie Matthes, Jonas Gross, Rebecca Waetzig, Tilman Heise, Selim Corbacioglu, Gunhild Sommer

**Affiliations:** 1Department of Pediatric Hematology, Oncology and Stem Cell Transplantation, University Hospital of Regensburg, Franz-Josef-Strauss Allee 11, 93053 Regensburg, Germany; 2Department of Internal Medicine 5—Hematology and Clinical Oncology, Ulmenweg 18, Friedrich Alexander University (FAU), 91054 Erlangen, Germany

**Keywords:** neuroblastoma, RIST, three-dimensional, cancer stem cells, CSC, 3D model, spheroids, RNA-binding protein, La (LARP3, La/SSB, La autoantigen)

## Abstract

**Simple Summary:**

A three-dimensional spheroid cell culture model resembles the architecture and cellular characteristics of solid tumors much more closely than the conventional monolayer cell cultures often applied in preclinical drug testing. In this study, we validate the efficacy of the RIST treatment protocol, a novel multimodal treatment regimen for high-risk neuroblastoma, in a neuroblastoma spheroid cell culture model characterized by an augmented neoplastic phenotype. The results also indicate the importance of the neuroblastoma spheroid model for preclinical drug testing in a rigorous, robust and efficient high-throughput format, which can be beneficial to identify more effective treatment options for children with high-risk neuroblastoma in the future.

**Abstract:**

Background: The outcome for patients with high-risk neuroblastoma remains poor and novel treatment strategies are urgently needed. The RIST protocol represents a novel metronomic and multimodal treatment strategy for high-risk neuroblastoma combining molecular-targeted drugs as ‘pre-treatment’ with a conventional chemotherapy backbone, currently evaluated in a phase II clinical trial. For preclinical drug testing, cancer cell growth as spheroid compared to mo-nolayer cultures is of advantage since it reproduces a wide range of tumor characteristics, including the three-dimensional architecture and cancer stem cell (CSC) properties. The objective of this study was to establish a neuroblastoma spheroid model for the rigorous assessment of the RIST treatment protocol. Methods: Evaluation of CSC marker expression was performed by mRNA and protein analysis and spheroid viability by luminescence-based assays. Aberrant expression of RNA-binding protein La in neuroblastoma was assessed by tissue microarray analysis and patients’ data mining. Results: Spheroid cultures showed increased expression of a subgroup of CSC-like markers (CXCR4, NANOG and BMI) and higher Thr389 phosphorylation of the neuroblastoma-associated RNA-binding protein La when compared to monolayer cultures. Molecular-targeted ‘pre-treatment’ of spheroids decreased neoplastic signaling and CSC marker expression. Conclusions: The RIST treatment protocol efficiently reduced the viability of neuroblastoma spheroids characterized by advanced CSC properties.

## 1. Introduction

As the most common extracranial solid tumor in childhood [1], neuroblastoma (NB) accounts for around 15% of all pediatric cancer deaths [2]. Derived from embryonic neural crest cells, the NB tumors typically arise along the sympathetic nervous system [1,3,4]. NB exhibits diverse clinical behavior, from spontaneous remission [5] to a widespread metastatic disease with poor outcomes and event-free survival rates below 50% [2,6,7]. Despite intensive treatment regimens, the prognosis for patients with high-risk relapsed or treatment-refractory NB remains poor and novel treatment strategies are urgently needed [8,9]. The pre-treatment risk stratification for this cancer relies on the International Neuroblastoma Risk Group (INRG) classification system, including age at diagnosis, histologic category, grade of tumor differentiation, tumor cell ploidy, 11q aberration and status of the oncogene MycN [2,10,11]. The genomic amplification of the oncogene MycN is currently the most important prognostic marker of poor outcomes, rapid tumor progression and treatment failure [2,3,12,13].

The RIST therapy, a novel multimodal treatment design for high-risk NB, is based on metronomically combining molecular-targeted biologicals, mTOR inhibitor Rapamycin and tyrosine-kinase inhibitor Dasatinib, with a conventional well-established chemotherapeutic backbone, consisting of the topoisomerase inhibitor Irinotecan and alkylator Temozolomide [14]. The mechanism of action of this metronomic approach is to target oncogenic and survival pathways with the molecular-targeted ‘pre-treatment’. This repetitively applied sequence induces predominantly a cell-cycle-synchronizing and chemo-sensitizing effect, catalyzing synergistically DNA damage induced by the consecutively applied chemotherapy [15]. Thus, this combination therapy uses drug combinations in a synergistic or additive manner, enhancing the therapeutic efficacy by applying lower dosages with reduced toxicity to non-cancerous cells compared to monotherapy [15]. Combining the benefits of combination therapy with metronomic scheduling, and frequently administering therapeutic agents at low dosages, toxicity is minimized [16,17] and the onset of drug resistance effectively delayed by providing more sustained apoptosis [18,19]. The results obtained from the phase II prospective randomized RIST clinical trial (ClinicalTrials.gov Identifier: NCT01467986) are currently under evaluation [14].

The majority of in vitro drug testing on cancer cells is performed on conventional two-dimensional (2D) cell cultures [20]. Nonetheless, monolayer cell growth does not fully represent the in vivo tumor biology, architecture and microenvironment, considering their lack of metabolic gradients and cell polarity, limiting their capability in accurately testing new therapeutic agents [20,21,22]. In contrast to 2D cell cultures, cancer cells grown in three-dimensional (3D) spheroid cultures reproduce a wide range of structural, physiological and biological features of in vivo avascular solid tumors, as well as gene expression profiles and cancer stem cell (CSC)-like properties important to the tumor’s drug response and resistance profile [21,23]. The spontaneous assembly of cells forming spheroids on ultra-low attachment (ULA) plates, not requiring any additional external biomaterials, remains one of the most established models for 3D cell culture due to its simplicity and reproducibility [21,23].

Cancer stem cells (CSCs) are known to have high tumorigenic potential, combining self-renewal, pluripotency and proliferating capacity with resistance to treatments such as conventional chemotherapy and radiation [24,25,26,27]. Therefore, CSCs are very likely involved in heterogenic tumor differentiation and progression, leading to metastasis, tumor recurrence and drug resistance [25,27]. Established CSC markers are expressed in various types of solid tumors, including neuroblastoma, such as chemokine receptor CXCR4, associated with migration, metastasis and tumor growth [28,29,30,31,32]; pluripotency-driving transcription factor NANOG, correlating with metastasis and poor prognosis [28,33]; as well as self-renewal factor BMI1, involved in tumorigenesis and chemoresistance [27,29]. Targeting the CSC population in neuroblastoma is essential to identify new and more effective therapeutic options, especially for high-risk patients with high resistance profiles, and relapsed or treatment-refractory tumors [27,34].

The aim of this study was to assess the RIST treatment protocol in a neuroblastoma spheroid culture model. Optimal conditions for spheroid growth on 96-well ultra-low-attachment (ULA) plates for three MycN-amplified (MNA) (SK-N-BE(2), Kelly, IMR-32) and two MycN-non-amplified (MNN) (SK-N-AS, SK-N-SH) NB cell lines were established. The 3D cultures were defined by measuring the size and viability of spheroids, as well as the expression of three CSC-like markers (CXCR4, NANOG, BMI1). Our results demonstrated the effective inhibition of neoplastic signaling and reduced expression of a subgroup of CSC-like markers after molecular-targeted ‘pre-treatment’ in spheroids of all NB cell lines tested.

## 2. Materials and Methods

Cell culture: Neuroblastoma and Ewing’s sarcoma cell lines were originally purchased from either the American Type Culture Collection, ATCC (SK-N-SH, SK-N-AS, SK-N-MC), or from the Deutsche Sammlung von Mikroorganismen und Zellkulturen, DSMZ (SK-N-BE(2), IMR-32, Kelly). Cells were cultivated at 37 °C in a 5% CO_2_ atmosphere. The SK-N-SH, SK-N-AS and SK-N-BE(2) lines were cultured in 1:1 EMEM (Lonza, #12-125F)/Ham’s F12 (Biochrom, Schaffhausen, Switzerland, #F0815), supplemented with 10% heat-inactivated fetal bovine serum (FBS, Sigma, #F7524), 1% penicillin/streptomycin (Sigma, #P0781) and 2 mM L-glutamine (Gibco Life Technologies, #25030081). The cell line IMR-32 was cultured in RPMI 1640 (Gibco Life Technologies, #31870-025) with 20% heat-inactivated FBS, 1% penicillin/streptomycin and 1% non-essential amino acids (MEM NEAA, Gibco Life Technologies, #11140-035). The cell line Kelly was cultured in RPMI 1640 supplemented with 10% heat-inactivated FBS, 1% penicillin/streptomycin and 2 mM L-glutamine. The cell line SK-N-MC was cultured in DMEM (Gibco Life Technologies, #21969-035) supplemented with 10% heat-inactivated FBS, 1% penicillin/streptomycin, 1% MEM NEAA and 2 mM L-glutamine. All cell lines were regularly tested for mycoplasma contamination. For 3D cultures, ultra-low-attachment (ULA) 96-well plates with round bottoms (Corning, #7007) were used. FBS was substituted for 5% knock-out serum replacement (KOSR, Gibco Life Technologies, #A31815-01) [28] in SK-N-SH, SK-N-AS, SK-N-BE(2) and IMR-32, enhancing stem cell like-properties and preventing differentiation due to its pro-survival activity [28,35,36].

Drug treatment: Temozolomide (BioVision, #2226-19), SN-38, the active metabolite of Irinotecan (Tocris, #2684), Dasatinib (#D-3307) and Rapamycin (#R-5000), both purchased from LC Laboratories, were dissolved in 100% dimethyl sulfoxide (DMSO, Sigma, #D2650) as a vehicle and stored at −20 °C. All drugs were diluted in cell-line-appropriate media to the desired concentrations immediately before treatment and applied at the indicated time points. Appropriate DMSO concentrations <1% final concentration were used as a control. Drug concentrations for the combined treatments are listed in Table 2. A scheme of the experimental design to evaluate the impact of RIST treatment on spheroid viability is displayed in Figure 4.

Viability analysis: To determine the IC_50_ concentration, the CellTiter-Glo^®^ 3D Cell Viability Assay (Promega, #G9681), based on measuring an ATP-dependent luminescent signal, was performed according to the manufacturer’s instructions [37]. The luminescence was recorded in a CLARIOStar^®^ Plate Reader and analyzed with the CLARIOStar^®^ MARS analysis software. Cell viability was calculated by normalizing the measured viability to DMSO (vehicle) controls. CellTox Green Dye (Promega, #G8741) was applied according to the manufacturer’s instructions. Three replicates were used for each drug combination and experiments were independently repeated at least three times.

RNA isolation and cDNA synthesis: Total RNA from NB spheroids was isolated using the RNeasy Mini Kit (Qiagen, #74106), following the manufacturer’s instructions. For purifying the isolated RNA, DNase digestion with DNase I was performed using the RNAse-free DNase Set (Qiagen, #79254). Quantification and purity were measured with the ND spectrophotometer NanoDrop. Complementary DNA (cDNA) was reverse-transcribed using the LunaScript RT Supermix (New England Biolabs, #E3010), according to the manufacturer’s instructions, with ‘No RT’ controls to verify the absence of genomic DNA in each template and therefore prevent quantification interference during qPCR.

Quantitative PCR (qPCR): To test for CSC marker expression, qPCR was performed using the SYBR Green Master mix (Roche, #04887352001) on a thermocycler system (Analytik Jena). PCR product quantification was measured by the relative quantification method of Livak and Schmittgen [38], normalized to the reference gene GAPDH and indicated as a fold change in CSC marker expression between cells grown in 2D monolayer versus 3D spheroid cell cultures. Primers used were CXCR4 (#QT00223188), NANOG (#QT01844808) and GAPDH (#QT00079247), all QuantiTect Primer Assays (Qiagen), as well as BMI1 (#PPH57778A), RT^2^qPCR Primer Assay (Qiagen).

Immunoblot analysis: Cells were lysed and protein was analyzed by SDS-polyacrylamide gel electrophoresis (SDS-PAGE). The following antibodies were applied: 4E-BP1 (Cell Signaling, #9644), phospho-4E-BP1(Ser65, Cell Signaling, #9456), MycN (Cell Signaling, #9405), p70S6 kinase (Cell Signaling, #9202), phospho-p70S6 kinase (Thr389, Cell Signaling, #9234), Src kinase (Cell Signaling, #2108), phospho-Src (Tyr416, Cell Signaling, #2101), GAPDH (Santa Cruz, #25778), CXCR4 (Abcam, #124824), BMI1 (Cell Signaling, #5856), NANOG (Cell Signaling, #4903), La protein (antibody 3B9 [39]) and phosho-La protein [40]. As secondary antibodies, horseradish peroxidase conjugated (Dianova) were applied. ImageJ 1.53a (National Institutes of Health, Bethesda, MD, USA) was used for quantification of the protein expression.

Immunohistochemistry: The immunohistochemical staining was performed applying the monoclonal human La-specific 3B9 antibody, as described in detail in [41,42], on a blastoma and normal tissue microarray (TMA: MC809, Biomax). For scoring of the predominantly nuclear staining of the La protein, five noncoincident microscopic fields were consecutively analyzed, 100 cells were counted per field, the total positive cells and negative cells from 5 fields were summed and a percentage was calculated. Results were depicted as percentages based on the largest population of benign cells present in normal tissue with respect to tumor cells present in cancer tissue. 

Statistical analysis: All experiments were performed at least three times. The results were stated as means of at least three technical replicates ± standard deviation (SD). For microscope image analysis, ImageJ 1.53a (National Institutes of Health, Bethesda, MD, USA) was used. Statistical data analysis was performed on GraphPad Prism 9 using the unpaired Student’s *t*-test. *p* values < 0.05 were considered statistically significant. Three degrees of significance were determined: * < 0.05, ** < 0.01 and *** < 0.001.

## 3. Results

### 3.1. Establishment of Culture Conditions for NB Spheroids

To define optimal spheroid culture conditions for various NB cell lines, we seeded increasing cell numbers from 10 to 1000 cells per well of two MycN-non-amplified (MNN) NB cell lines (SK-N-SH and SK-N-AS) in round-bottom ULA 96-well plates and monitored spheroid growth over time by taking phase-contrast images (Figure 1a); we determined the sizes of NB spheroids by measuring the diameters after 3, 7, 10 and 14 days (Figure 1b). After 7 days of culture, as few as 10 cells per 96-well gave rise to small spheroids (50 µm in diameter); however, seeding a cell number of 1000 cells per 96-well increased considerably the size of spheroids for both cell lines to more than 200 µm in diameter after 7 days in culture (Figure 1b). Whereas SK-N-SH cells formed compact, round spheroids, SK-N-AS spheroids appeared round and slightly puffy-edged (Figure 1a).

In addition, we monitored spheroid growth for three MycN-amplified (MNA) cell lines (SK-N-BE(2), IMR-32, Kelly) by plating again increasing cell numbers in round-bottom ULA 96-well plates and monitoring the phenotype and size of spheroids over time. After 7 days, SK-N-BE(2) and IMR-32 formed compact, round spheroids, whereas Kelly formed spheroids with less defined edges (Appendix A). Furthermore, IMR-32 and Kelly cells formed overall smaller spheroids compared to SK-N-BE(2) when comparing cell numbers seeded per ULA 96-well plate over time (Appendix A).

Table 1 summarizes the established culture conditions for optimal spheroid growth of all five NB cell lines tested. The NB cell line SK-N-BE(2) formed the largest spheroids after day 3 and day 7 in culture, even though the lowest cell number (500 cells per 96-well) was seeded (Table 1). The cell lines SK-N-SH, SK-N-AS and IMR-32 formed spheroids of similar size and smaller in diameter when compared to SK-N-BE(2) and Kelly (Figure 2). In contrast to all other NB cell lines tested, Kelly cells needed 10% FBS as a culture supplement instead of 5% KOSR and showed the least compact and round spheroid phenotype (Appendix A) with an enlarged dead cell population (Appendix A).

### 3.2. Increased mRNA Expression of CSC-Like Markers in NB Spheroids Compared to Monolayer Cultures

Testing whether NB spheroids gained cancer stem cell (CSC) properties, we determined the mRNA expression levels of CSC-like markers in cells grown as spheroids in 3D culture compared 2D monolayer culture. Hence, for all five NB cell lines, we isolated RNA from both monolayer and spheroid cultures and quantified the mRNA expression of CSC markers CXCR4, NANOG and BMI1 by reverse transcription followed by quantitative PCR. In all NB cell lines tested, the CXCR4 mRNA expression was increased in spheroid compared to monolayer cultures. In MNN cell lines, CXCR4 mRNA expression was dramatically upregulated (10- to 18-fold), whereas, in MNA cell lines, the increase was 3- to 5-fold in spheroid cultures (Figure 3a). Furthermore, the expression of NANOG and BMI1 mRNA was elevated 2- to 3-fold in MNN spheroids compared to monolayer cultures; however, for the MNA cell lines, only IMR-32 spheroids showed a two-fold increase in BMI1 mRNA expression (Figure 3b,c).

Taken together, these data indicate a strong increase in CXCR4 mRNA expression in NB spheroids compared to monolayer cultures for all five NB cell lines tested. Interestingly, the effect was noticeably pronounced in NB spheroid cultures of MNN when compared to MNA cell lines. Furthermore, NANOG and BMI1 mRNA expression was also, although to a lesser extent, augmented in spheroids compared to monolayer cultures in both MNN cell lines (SK-N-SH and SK-N-AS) and BMI1 mRNA expression in MNA cell line IMR-32.

### 3.3. Reduced Viability of NB Spheroids after ‘Pre-Treatment’ with Rapamycin plus Dasatinib

The novel multimodal RIST treatment protocol combines two molecular targeted drugs, the mTOR inhibitor Rapamycin and the tyrosine-kinase inhibitor Dasatinib, as a ‘pre-treatment’, with conventional chemotherapeutics, the topoisomerase inhibitor Irinotecan and the alkylating agent Temozolomide (Figure 4). Previous studies in monolayer cultures showed that ‘pre-treatment’ with Rapamycin plus Dasatinib (R+D) synchronizes NB cells in the cell cycle G1 phase and chemosensitizes them towards conventional ‘chemo-treatment’ with Irinotecan plus Temozolomide (I+T) [43].

Here, we tested whether the R+D ‘pre-treatment’ affects the viability of NB spheroids. Therefore, we seeded NB cells in round-bottom ULA 96-well plates and treated them with R+D at the concentrations depicted in Table 2. The effective concentrations of the R+D ‘pre-treatment’ and I+T ‘chemo-treatment’ are depicted in Table 2 and were previously determined for NB cell lines in monolayer cultures by assessing the half maximal inhibitory concentrations (IC_50_) for single-drug treatment applying viability assays, followed by determining the synergistic effects of combination treatments by using the Chou–Talalay method [43,44,45]. Due to synergistic effects, in combination treatments (R+D, I+T or RIST), all drugs were used below their IC_50_ values applied for single-drug treatments [43,44].

To test whether the R+D ‘pre-treatment’ affected the viability of NB spheroids, cells were seeded in round-bottom ULA 96-well plates according to seeding numbers, as depicted in Table 1. The viability of R+D- and control (vehicle)-treated NB spheroids was assessed after 72 h by applying the CellTiter-Glo 3D Cell Viability Assay (Figure 4). The results showed that the R+D ‘pre-treatment’ reduced the viability of NB spheroids by 40–60% compared to the control treatment for all tested NB cell lines (Figure 5). These results correspond well with previous studies in NB monolayer cultures showing approximately 50% reduced viability after R+D ‘pre-treatment’ when applying the drug concentrations listed in Table 2 [43,44].

### 3.4. Neoplastic Signaling Response to R+D ‘Pre-Treatment’ with Rapamycin plus Dasatinib in NB Monolayer and Spheroid Cultures

To determine how R+D ‘pre-treatment’ with Rapamycin plus Dasatinib (R+D) affects the neoplastic signaling response, the mammalian Target of Rapamycin (mTOR) signaling pathway and phosphorylation of the Src kinase family were assessed in NB monolayer and spheroid cultures. Herein, we analyzed the impact of Rapamycin on the phosphorylation status of downstream targets of mTOR complex 1 (mTORC1), such as the phosphorylation of ribosomal protein S6 kinase (p70S6K) at threonine 389 (Thr389) and the phosphorylation of translation initiation factor 4E-BP at serine 65 (Ser65). Furthermore, changes in the phosphorylation status of Src kinase at tyrosine 416 (Tyr416) upon tyrosine-kinase inhibitor Dasatinib treatment were analyzed. We treated monolayer and spheroid cultures of the MNN cell line SK-N-AS and the MNA cell line SK-N-BE(2) with R+D or with vehicle (control) at the concentrations depicted in Table 2. After 24 h, cell lysates were assessed by immunoblot analysis, demonstrating that the R+D ‘pre-treatment’ efficiently inhibited both the mTOR signaling pathway and Src kinase activation in monolayers as well as in spheroid cultures for both NB cell lines (Figure 6). 

### 3.5. R+D ‘Pre-Treatment’ Reduces the Elevated Protein Expression of CXCR4, BMI1 and NANOG in NB Spheroids Compared to Monolayer Cultures

Our above-reported results (Figure 3) revealed an mRNA increase for the CSC-like markers CXCR4, BMI1 and NANOG in spheroid compared to monolayer cultures in NB cell lines tested. Immunoblot analysis of monolayer cultures demonstrated also higher protein expression of CXCR4 in MNA (Kelly, SK-N-BE(2)) compared to MNN (SK-N-AS, SK-N-SH) cell lines, whereas the protein expression of BMI1 and NANOG did not correlate with the MycN status of the NB cell lines (Figure 7a).

Notably, in neuroblastoma cell lines SK-N-AS and SK-N-BE(2), the protein expression of CXCR4, BMI1 and NANOG was raised in spheroid compared to monolayer cultures, suggesting an augmented cancer stemness phenotype expressed under 3D culture conditions (Figure 7b). Most importantly, R+D ‘pre-treatment’ compared to control (vehicle) treatment proficiently reduced the elevated CXCR4, BMI1 and NANOG protein expression in spheroid cultures in both neuroblastoma cell lines (Figure 7b). 

### 3.6. The RIST Treatment Protocol Affects the Viability of NB Spheroids 

To test whether the RIST treatment affects the viability of NB spheroids, we seeded five different NB cell lines in round-bottom ULA 96-well plates and treated them on the same day with the combination of Rapamycin plus Dasatinib (R+D) at the concentrations depicted in Table 2. After 72 h of R+D ‘pre-treatment’ (Figure 4), 90% of the medium was changed, leaving the spheroids untouched in the remaining medium. Fresh medium was added and the integrity of spheroids was microscopically monitored after the change. Twenty-four hours later, the chemotherapeutic treatment with Irinotecan plus Temozolomide (I+T) was added for 72 h at the concentrations outlined in Table 2. The viability of NB spheroids was assessed by applying the CellTiter-Glo 3D Cell Viability Assay. The multimodal RIST treatment significantly reduced the viability of spheroids in all NB cell lines tested below 20%, except for SK-N-BE(2) spheroids, which responded less (Figure 8).

### 3.7. Overexpression of the RNA-Binding Protein La Correlates with Low Survival in Neuroblastoma 

The RNA-binding protein La is overexpressed in a variety of cancer entities and has been shown to play a critical role in cancer pathobiology [46]. Immunohistochemical analysis of normal and cancer tissue applying the commercially available tissue microarray TMA MC809, followed by the scoring of cells positively stained for the La protein, indicated the overexpression of the La protein in neuroblastoma, nephroblastoma and glioblastoma tissue compared to normal cerebrum, cerebellum, and kidney tissue (Figure 9a,b). 

The analysis of patients’ data retrieved from the R2 website (R2: Genomics Analysis and Visualization Platform (https://r2.amc.nl, dataset ID: GSE62564, accessed on 9 March 2023) revealed a significant correlation between high La mRNA expression and the reduced overall survival probability of neuroblastoma patients (Figure 9c).

### 3.8. Increased Phosphorylation of RNA-Binding Protein La at Thr389 in NB Spheroids Can Be Reduced by R+D ‘Pre-Treatment’ 

Recently, we have shown that cancer-associated and aberrantly overexpressed La protein is critical for transforming growth factor beta (TGFβ)-induced epithelial to mesenchymal transition (EMT) and the gain of CSC properties in head and neck cancer cells [40]. The TGFβ-induced EMT and the increased CSC phenotype correlated with the AKT-mediated phosphorylation of La at threonine 389 (Thr389), suggesting a role of Thr389 phosphorylation in cancer cell plasticity and stemness. 

Immunoblot analysis revealed a slight increase in the Thr389 phosphorylation of La in MNA neuroblastoma cell lines (Kelly, IMR-32, SK-N-BE(2)) when compared to MNN neuroblastoma (SK-N-AS, SK-N-SH) and Ewing‘s sarcoma (SK-N-MC) cell lines (Figure 10a). Comparing monolayer with spheroid cultures, the Thr389 phosphorylation of La was slightly increased in spheroids of MNN cell line SK-N-AS and strongly augmented in spheroids of MNA cell line SK-N-BE(2) (Figure 10b). Notably, the R+D ‘pre-treatment’ significantly reduced the Thr389 phosphorylation of La in spheroids in both NB cell lines. Taken together, our data also indicate a novel correlation between the Thr389 phosphorylation of La and the reduced viability of R+D-treated NB spheroids.

## 4. Discussion

New treatment strategies are urgently needed for high-risk children with relapsed or treatment-refractory neuroblastoma. Targeting relevant signaling pathways—such as the neoplastic PI3K/Akt/mTOR pathway—revealed promising efficacy in monolayer NB cell cultures, and corresponding therapeutic agents were tested in various clinical trials [43]. However, these promising preclinical results could often not be translated into the clinic due to a poor correlation between their efficacy in two-dimensional monolayer cell culture models and their success rates in patients [20,48,49]. One option to reduce these limitations is to culture cancer cells as three-dimensional spheroids, more closely resembling the in vivo tumor biology and architecture, accompanied by significantly increasing malignant cancer cell phenotypes and the presence of CSCs explaining lower sensitivities to therapeutic agents tested in monolayer cell cultures [20,22,50,51,52]. Furthermore, spheroid models not only decrease the discrepancy between in vivo tumor tissue and in vitro cell culture but were also established to reduce the use of animal models [20]. 

In the present study, we established a spheroid cell culture model for neuroblastoma. Compared to monolayer cultures, NB spheroids were characterized not only by significantly higher expression of a subgroup of CSC markers, including CXCR4, NANOG and BMI1, but also increased T389 phosphorylation of cancer-associated RNA-binding protein La. Nevertheless, treatment with the multimodal RIST protocol efficiently inhibited neoplastic signaling, viability and CSC-like marker expression in NB spheroids. The results indicate the efficiency of the RIST treatment protocol and, in addition, emphasize the NB spheroid model as favorable compared to monolayer cultures for the preclinical testing of new drugs in a rigorous and efficient screening format, which can be beneficial to identify more effective treatment options for children with high-risk neuroblastoma in the future.

MycN amplification is well established as an important predictor of poor outcomes in NB patients and defines 50% of the high-risk population [53]. Consequently, the present study focused on differentiation between MycN-non-amplified (MNN) and MycN-amplified (MNA) NB cell lines. However, additional non-overlapping groups of high-risk neuroblastoma, such as the groups of ‘alternative lengthening of telomeres (ALT)’ and of ‘telomerase gene (TERT) rearrangements’, are associated with a very poor prognosis [54]. In future studies, the RIST treatment response of representative NB cell lines should be evaluated.

Several studies have indicated that elevated CSC marker expression is induced by spheroid growth [21,22]. Comparing the CSC-like marker expression between the 2D and 3D growth of neuroblastoma cells, our results support this notion. Interestingly, in MycN-non-amplified (MNN) cell lines, the mRNA expression of all three CSC markers was strongly elevated in spheroid compared to monolayer cultures (10- to 18-fold for CXCR4) (Figure 3), whereas, in MycN-amplified (MNA) cell lines, the protein expression of CSC-like marker CXCR4 was already aberrantly elevated (Figure 7a) and only moderately increased (1.9-fold) between 2D and 3D culture conditions. These findings are in agreement with oncogene MycN driving a cancer stem-like phenotype, such as increased cell renewal, apoptotic resistance and metabolic flexibility [53]. 

Chemokine-receptor 4, CXCR4, after binding its ligand, stomal-derived factor 1, SDF1 (CXCL12), is important for the homing of hematopoietic stem cells and establishing a stem cell niche within the bone marrow [55]. In NB patients, high CXCR4 expression correlates with tumor cell migration and bone marrow metastasis, often described as ‘neuroblastoma metastatic cell homing’ [31,32,55,56,57]. CXCR4 has been previously suggested as a CSC-like marker in neuroblastoma [28], predominantly expressed in stage 3 and stage 4, compared to the more localized stages 1 and 2 [58]. Interestingly, the MycN-amplified oncogene status is linked to advanced stages of NB, as the upregulation of CXCR4 is associated with stages 3 and 4. Our results show that 3D compared to 2D growth significantly increased the expression of CXCR4 in all tested NB cell lines; however, in MNN cell lines, the fold increase in CXCR4 expression was three times higher compared to all three MNA cell lines tested (Figure 3a). The underlying cellular mechanism linking 3D growth with augmented CXCR4 expression in neuroblastoma cells is an open question. Interestingly, Ikegaki et al. demonstrated—next to other stemness factors and marker genes, such as NANOG, ABCG2 and LIN28—considerably enhanced CXCR4 levels in induced CSCs (iCSC) when compared to monolayer cultures of SK-N-AS [28]. In this study, the stemness of cancer cells was induced by treatment with the epigenetic modifier 5-aza-2′-deoxycytidine (5AdC), an inhibitor of DNA methylation, followed by long-term (>100 days) spheroid-forming culture conditions. Xenograft experiments demonstrated that the tumor-initiating capacity was much more potent when injecting iCSCs compared to monolayer NB cells into SCID mice [28]. This study suggests that 5AdC-induced epigenetic changes and the formation of an open chromatin structure are favorable for the enrichment of the CSC population in NB spheroid cultures.

The pluripotency-maintaining transcription factor NANOG maintains a high self-renewal capacity of embryonic stem cells and is a CSC-like marker associated with poor prognosis in neuroblastoma [28,29,33,59]. BMI1, on the other hand, as a member of the polycomb group (PcG) proteins, acts as an epigenetic silencer of tumor suppressor genes and is involved in tumor proliferation, progression and chemoresistance [60,61]. Due to its ability to foster a self-renewal capacity, BMI1 represents a CSC-like marker in NB [29,60] and is known to act as a cooperative oncogene with MycN [62]. Several publications indicate that there is a correlation between stemness and MycN amplification in NB, especially for the CSC-like markers NANOG and BMI1 [55,60,63]. In this context, Nardella et al. demonstrated a CSC phenotype in Lamin A/C-depleted NB cells (SH-SY5Y) in correlation with augmented NANOG and BMI1 expression [64]. The upregulation of NANOG and BMI1 expression was associated with increased MycN expression, as well as the elevated spheroid growth and tumor-initiating capacity of NB cells, in mice [64].

Our study demonstrates, for the first time, the aberrant overexpression of RNA-binding protein La in neuroblastoma. Furthermore, data mining revealed that high La mRNA expression correlates with poor overall survival in neuroblastoma patients. These findings are in agreement with several former reports demonstrating the aberrant overexpression of the RNA-binding protein La in a variety of cancer entities, including lung, cervical and head and neck cancer, as reviewed recently [46]. Cancer-associated La enhances tumor-promoting processes such as proliferation, migration, invasion, CSC marker expression and spheroid and tumor growth, and in addition contributes to the chemotherapeutic resistance of cancer cells [46]. Mechanistically, it has been suggested that the RNA chaperone activity of La supports, on one hand, the processing of non-coding RNAs and, on the other hand, the translation of selective mRNAs characterized by a highly structured 5′-untranslated region and often encoding tumor-promoting factors, such as cell cycle regulator cyclin D1 [65], tumor suppressor p53 regulator MDM2 [66] and anti-apoptotic factor Bcl2 [67]. Furthermore, a recent study emphasized the role of posttranslational modifications of La in cancer pathobiology. For example, the sumoylation of La [68] regulates the expression of the proliferation and angiogenesis-promoting factor STAT3 [69]. Furthermore, increased phosphorylation of La at threonine 389 (Thr389) correlates with TGFβ-induced epithelial-to-mesenchymal (EMT) transition and cancer cell plasticity [40] and spheroid viability (this study) and is—on the other hand—reduced by RIST treatment (this study). These findings support the notion that the cancer-associated RNA-binding protein acts as a non-oncogenic addiction factor, as reviewed recently [46], and that La phosphorylation at Thr389 is a novel molecular target for RIST therapy. 

An accurate analysis of cell viability and susceptibility to the given treatment is necessary to assess its preclinical efficacy and guide the development of new therapeutics with high precision. To decrease the discrepancy between in vivo tumor tissue and in vitro cell cultures, spheroid models were established. Data from previous studies report an increase in chemoresistance to therapeutic agents due to 3D spheroid growth compared to cancer cells grown as 2D monolayers [21,50,70]. Considering that the testing of treatment options on 3D models has higher predictive potential for chemoresistance, and our results showing significantly reduced viability, less expression of CSC-like markers and inhibited neoplastic signaling after the RIST treatment, the presented preclinical NB spheroid model validates the RIST treatment regimen as highly effective, even in an advanced tumor-like test setting.

## 5. Conclusions

The established neuroblastoma spheroid model—characterized by increased expression of a subgroup of CSC-like markers and increased Thr389 phosphorylation of the tumor-promoting RNA-binding protein La—was applied to assess the efficacy of the RIST treatment protocol, a novel multimodal treatment regimen for high-risk relapsed and refractory neuroblastoma, currently evaluated in a phase II clinical trial. Our results show the significant tumor-suppressive effect of the RIST treatment on cell viability and neoplastic signaling, emphasizing the importance of this efficient, reproducible, high-throughput 96-well 3D drug screening format to improve current treatment options for children with high-risk NB in the future.

## Figures and Tables

**Figure 1 cancers-15-01749-f001:**
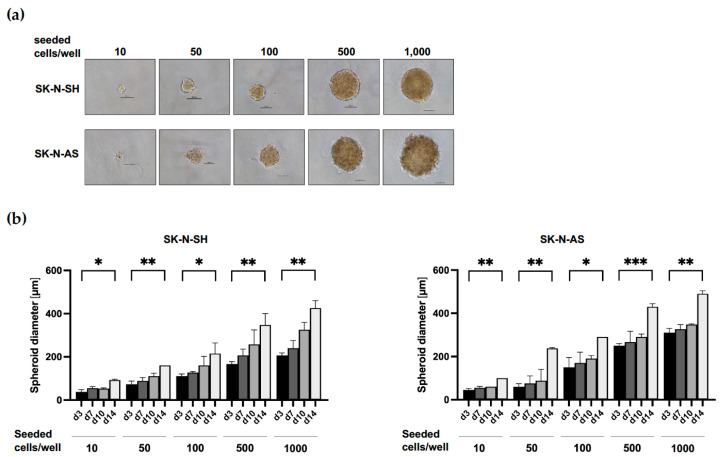
Establishment of spheroid culture conditions. (**a**) Phase-contrast imaging of SK-N-SH spheroids on day 7 and SK-N-AS spheroids on day 10 after seeding increasing cell numbers on round-bottom 96-well ULA plates. Scale = 100 µm. (**b**) Analysis of the diameter of SK-N-SH (**left**) and SK-N-AS (**right**) spheroids seeded with increasing cell numbers and grown on round-bottom 96-well ULA plates for 3 days (d3), 7 days (d7), 10 days (d10) and 14 days (d14). Three independent experiments (n = 3) in triplicate were performed. Degrees of significance (unpaired student’s *t*-test): * < 0.05, ** < 0.01 and *** < 0.001.

**Figure 2 cancers-15-01749-f002:**
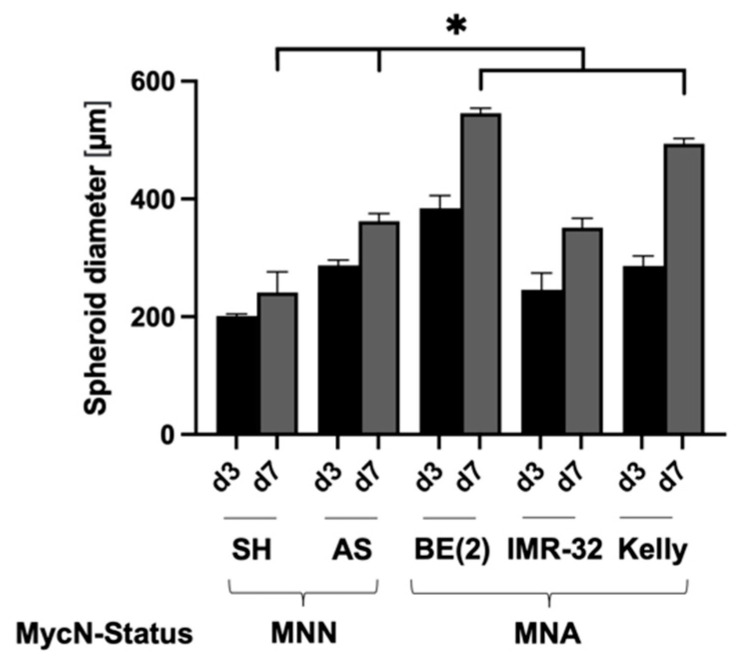
Size of spheroids of five different neuroblastoma cell lines grown in 3D culture. Diameter of spheroids from neuroblastoma cell lines SK-N-SH (SH), SK-N-AS (AS), SK-N-BE(2) (BE(2)), IMR-32 and Kelly grown under established conditions (Table 1) on 96-well ULA plates for 3 days (d3) or 7 days (d7). MycN-non-amplified (MNN), MycN-amplified (MNA). Three independent experiments (n = 3) in triplicate were performed. Degrees of significance (unpaired student’s *t*-test): * < 0.05.

**Figure 3 cancers-15-01749-f003:**
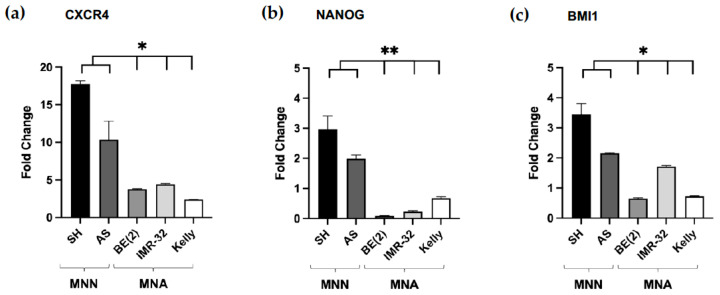
CSC-like marker expression in neuroblastoma spheroids compared to monolayer cell cultures. Fold change increase in cancer stem cell (CSC)-like marker (**a**) CXCR4, (**b**) NANOG and (**c**) BMI1 expression in spheroids compared to monolayer cultures of neuroblastoma cell lines SK-N-SH (SH), SK-N-AS (AS), SK-N-BE(2) (BE(2)), IMR-32 and Kelly determined by RTqPCR. Normalized to GAPDH expression. Statistical analysis of variance (ANOVA) was performed to determine the association between CSC-like marker expression in MNN and MNA cell lines. MycN-non-amplified (MNN), MycN-amplified (MNA). At least two independent experiments (n = 2) in triplicate were performed. Degrees of significance (unpaired student’s *t*-test): * < 0.05 and ** < 0.01.

**Figure 4 cancers-15-01749-f004:**
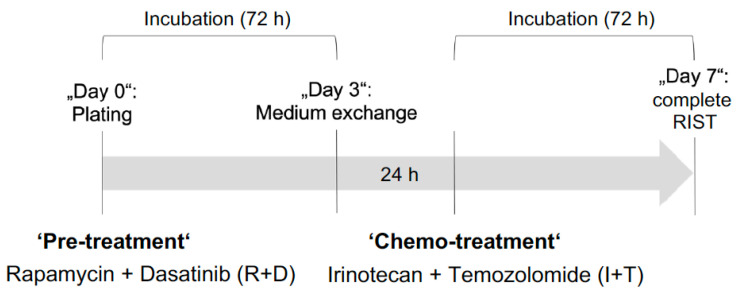
Scheme of the experimental design for the in vitro RIST treatment protocol.

**Figure 5 cancers-15-01749-f005:**
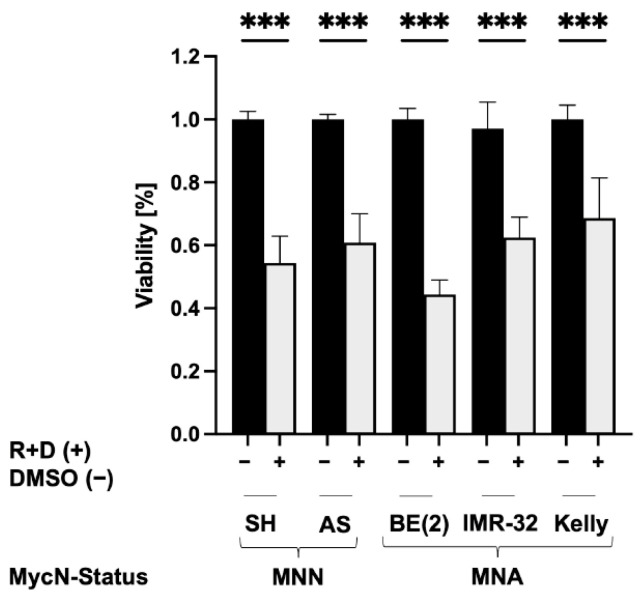
Viability of NB spheroids after R+D ‘pre-treatment’. To determine the impact of the Rapamycin plus Dasatinib (R+D) ‘pre-treatment’ on spheroid viability, spheroids were harvested on day 3 and analyzed applying the CellTiter-Glo 3D Cell Viability Assay. Viability of SK-N-SH (SH), SK-N-AS (AS), SK-N-BE(2) (BE(2)), IMR-32 and Kelly spheroids determined 3 days after treatment with R+D (+) or vehicle (−) DMSO treatment. MycN-non-amplified (MNN), MycN-amplified (MNA). Two independent experiments (n = 2) in triplicate were performed. Degrees of significance (unpaired student’s *t*-test): *** < 0.001.

**Figure 6 cancers-15-01749-f006:**
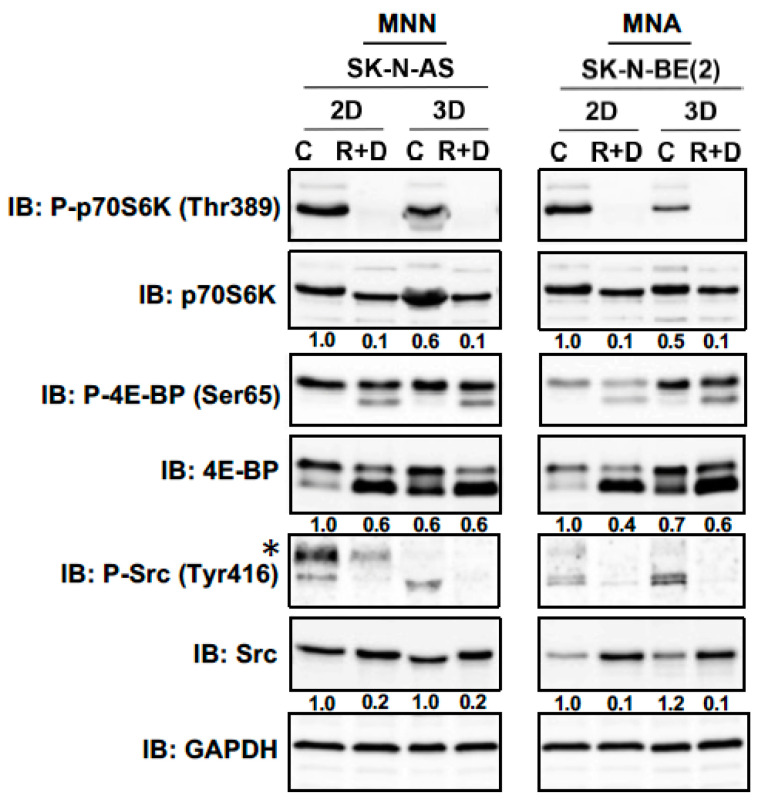
Neoplastic signaling pathway response to R+D ‘pre-treatment’ comparing 2D and 3D cultures. Immunoblot (IB) analysis showing the phosphorylation status of ribosomal protein S6 kinase (p70S6K), translation initiation factor 4E-BP and tyrosine kinase Src under control (C)- and Rapamycin plus Dasatinib (R+D)-treated conditions in two-dimensional (2D) monolayer and three-dimensional (3D) spheroid cultures of neuroblastoma cell line SK-N-AS and SK-N-BE(2). Numbers below bands state phosphoprotein (P) normalized to total protein expression. GAPDH was applied as loading control. C: Vehicle-treated control cells. R+D: Rapamycin plus Dasatinib treatment as depicted in Table 2. Star (*): unknown band. MycN-non-amplified (MNN), MycN-amplified (MNA). IB: Immunoblot. Original immunoblot (IB) images (Appendix A).

**Figure 7 cancers-15-01749-f007:**
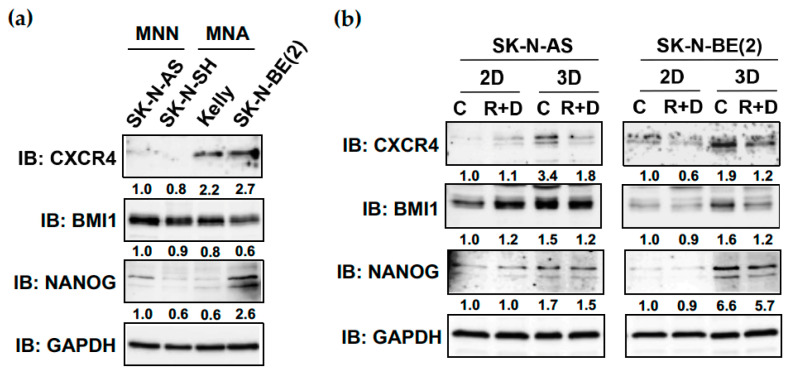
The R+D ‘pre-treatment’ reduces CXCR4, BMI1 and NANOG protein expression in NB spheroid cultures. Immunoblot (IB) analysis showing (**a**) CXCR4, BMI1 and NANOG protein expression in four different neuroblastoma cell lines grown in 2D cultures with medium supplemented with 10% FBS, and (**b**) CXCR4, BMI1 and NANOG protein expression under control (C)- and Rapamycin plus Dasatinib (R+D)-treated conditions comparing 2D and 3D cultures grown in medium supplemented with 5% KOSR. GAPDH was applied as loading control. Numbers below bands state protein expression normalized to GAPDH. C: Vehicle-treated control cells. R+D: Rapamycin plus Dasatinib treatment as depicted in Table 2. MycN-non-amplified (MNN), MycN-amplified (MNA). IB: Immunoblot. Original immunoblot (IB) images (Appendix A).

**Figure 8 cancers-15-01749-f008:**
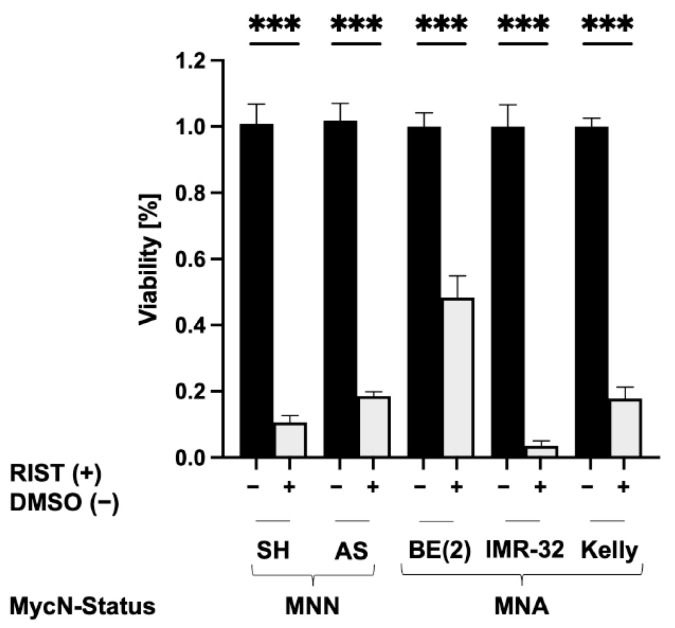
Viability of NB spheroids after RIST treatment. To determine the impact of the multimodal RIST treatment (Figure 4) on spheroid viability, spheroids were harvested on day 7 (complete RIST) and analyzed applying the CellTiter-Glo 3D Cell Viability Assay. NB cell lines: SK-N-SH (SH), SK-N-AS (AS), SK-N-BE(2) (BE(2)), IMR-32 and Kelly. RIST (+) or DMSO (−) (vehicle) treatment. MycN-non-amplified (MNN), MycN-amplified (MNA). At least three independent experiments (n = 3) in triplicate were performed. Degrees of significance (unpaired student’s *t*-test): *** < 0.001.

**Figure 9 cancers-15-01749-f009:**
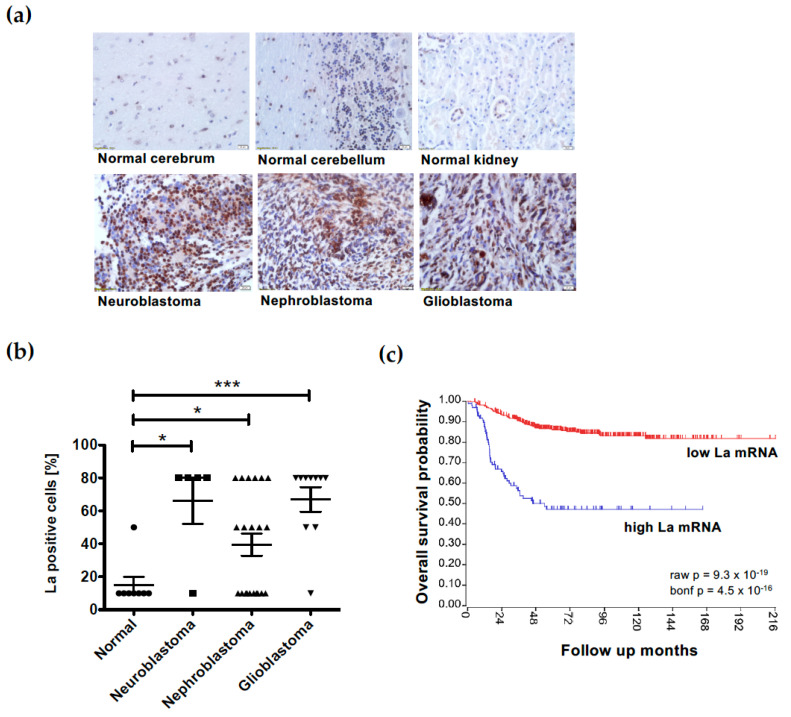
High expression of the RNA-binding protein La correlates with low overall survival probability in neuroblastoma. (**a**,**b**) Immunohistochemical analysis of tissue microarray TMA MC809 demonstrates significant overexpression of the La protein in neuroblastoma, nephroblastoma and glioblastoma tissue compared to normal cerebrum, cerebellum and kidney tissue. (**c**) Kaplan–Meier curve indicating that elevated expression of La mRNA is an indicator for reduced overall survival probability in neuroblastoma. The neuroblastoma data set SEQC-498 consists of 498 cases including 398 patient samples with low (red) La and 100 with high (blue) La mRNA expression levels, leading to a significant difference. The datasets for the Kaplan–Meier curves were retrieved from the R2 platform (https://r2.amc.nl, dataset ID: GSE62564, accessed on 9 March 2023). Degrees of significance (unpaired student’s *t*-test): * < 0.05 and *** < 0.001.

**Figure 10 cancers-15-01749-f010:**
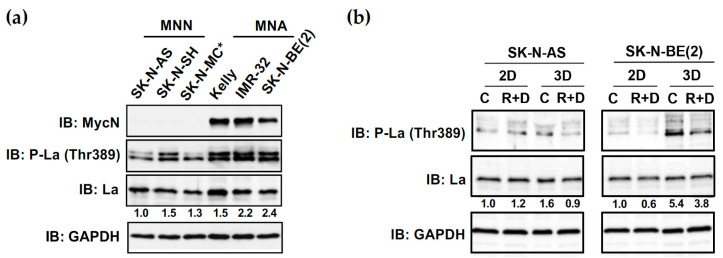
R+D ‘pre-treatment’ reduced augmented Thr389 phosphorylation of RNA-binding protein La in NB spheroids. Immunoblot (IB) analysis indicates (**a**) the expression level of oncogene MycN and the La protein as well as the phosphorylation status of La at threonine 389 (Thr389) in various neuroblastoma cell lines (SK-N-AS, SK-N-SH, Kelly, IMR-32, SK-N-BE(2)) and SK-N-MC*. grown in 2D monolayer cultures in medium supplemented with 10% FBS. The star (*) indicates that SK-N-MC has been originally described as neuroblastoma but is now widely regarded as an Askin‘s tumor cell line related to Ewing‘s sarcoma [47]. In (**b**), the phosphorylation status of La at Thr389 is depicted, comparing 2D and 3D cultures under control (vehicle)- and Rapamycin plus Dasatinib (R+D)-treated conditions. Cells were grown in medium supplemented with 5% KOSR. Numbers below bands state phosphoprotein (P-La (Thr389)) normalized to total La protein expression. GAPDH was applied as loading control. C: Vehicle-treated control cells. R+D: Rapamycin plus Dasatinib treatment as depicted in Table 2. MycN-non-amplified (MNN), MycN-amplified (MNA). IB: Immunoblot. Original immunoblot (IB) images (Appendix A).

**Table 1 cancers-15-01749-t001:** Summary of optimal 3D culture conditions for five neuroblastoma cell lines and average spheroid size (diameter) after 3 days (d3) and 7 days (d7) of culture growth.

Cell Line	MycN Status	Supplement	Cells/Well	Size d3 [µm]	Size d7 [µm]
SK-N-SH	MNN	5% KOSR	1000	201 ± 2.9	241 ± 7.5
SK-N-AS	MNN	5% KOSR	1000	287 ± 7.8	361 ± 16.5
SK-N-BE(2)	MNA	5% KOSR	500	384 ± 15.3	546 ± 5.9
IMR-32	MNA	5% KOSR	1000	246 ± 7.8	351 ± 11.2
Kelly	MNA	10% FBS	1000	286 ± 15.8	494 ± 7.6

MNN: MycN-non-amplified; MNA: MycN-amplified; KOSR: knock-out serum replacement; FBS: fetal bovine serum.

**Table 2 cancers-15-01749-t002:** Drug concentration applied for the R+D ‘pre-treatment’ and I+T ‘chemo-treatment’ in the multimodal in vitro RIST treatment protocol.

Cell Line	MycN Status	Rapamycin [µM]	Dasatinib [µM]	Irinotecan [nM]	Temozolomide [µM]
SK-N-SH	MNN	0.20	0.10	2.00	150.00
SK-N-AS	MNN	0.10	0.10	4.44	133.33
SK-N-BE(2)	MNA	5.00	20.00	1.00	225.00
IMR-32	MNA	0.13	0.05	0.40	120.00
Kelly	MNA	2.00	5.00	1.00	225.00

MNN: MycN-non-amplified; MNA: MycN-amplified.

## Data Availability

The datasets for the Kaplan–Meier curves were retrieved from the R2 platform (https://r2.amc.nl, dataset ID: GSE62564 accessed on 9 March 2023).

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
