# Peer review of "Evaluating the RIST Molecular-Targeted Regimen in a Three-Dimensional Neuroblastoma Spheroid Cell Culture Model"

_cancers, 2023, doi:10.3390/cancers15061749_

Round 1

Reviewer 1 Report

Evaluating the RIST molecular-targeted regimen in a three-dimensional neuroblastoma spheroid cell culture model 

The objective of this study was to establish a neuroblastoma spheroid model for rigorous assessment of the RIST treatment protocol. 

Evaluation of CSC marker expression by mRNA and protein analysis and spheroid viability by luminescence-based assays. This sentence does not have a verb.

What is the significance of CSC markers increasing in spheroid models? Does this mean they more faithfully recapitulate a tumour in vivo?

Derived from primitive undifferentiated neuroectodermal cells NB tumours typically arise along the sympathetic nervous system, is this neural crest rather?

The authors could define risk stratification in this cancer.

MYCN amplification is the strongest predictor of poor outcomes seen in nearly 50% of high-risk cases. Also, ALT phenotype and TERT rearrangements are important in high-risk cases.

RIST therapy has been mentioned but the authors do not mention if there are certain eligibility criteria within the high-risk group to qualify for this treatment (any particular molecular characteristics such as upregulated mTOR etc?).

CSCs usually have markers such as CD44 and CD133. CSCs in neuroblastoma are a matter of controversy, the van Groningen showed that CD133 positive and negative cells could propagate tumours in vivo xenograft models (doi:10.1038/ng.3899). So whether CXCR4, BMI1 and NANOG are definitive markers of these stem cells in neuroblastoma is far from “established”. Also, CSCs, propagate the existing tumour in vivo rather than establishing a new tumour (tumour-initiating cells), and resisting therapy are property of tumour persister cells rather than CSCs or tumour-initiating cells, so the papers cited in lines 26-30 may not be specifically making reference to CSCs (unfortunately, these three cell types get used interchangeably across the literature).  Also, the Bahmad review paper cited by the authors lists a long list of potential CSCs in NB (DLK1, CD114, BMI1, CD44, CD133, C-KIT, CD24, FZD6, ALDH1, LGR5, TLX, ABCG2, Nestin, JARDI1B, SPDYA, TRPM7, Lamin A/C and Cam L1). I would therefore recommend using these terms more tentatively and less matter-of-fact.

What links CSCs to high-risk NB?  Since we are still not sure what the markers of CSCs, how could we establish whether they are enriched in high-risk tumours compared to low-risk (and thus make it a reliable model)? 

The authors could make the claims more general because the neoplastic potential and a subgroup of CSC markers were reduced.

In figure 1, did the authors do any stats on the seeding densities and spheroid diameters?

In figure 2, The cell lines SK-N-SH, SK-N-AS and IMR-32 formed spheroids of similar size and smaller in diameter when compared to SK-N-BE(2) and Kelly (Fig. 2). What does this signify given that IM-32 (MYCN amplified: MA) is grouped similarly compared to the non-MA cells? Sphere diameter is not necessarily linked to MYCN status.

In figure 3 it is interesting that the MA cell lines have much lower CSC markers than the non-MA. What does this signify?

In table 2 the authors mention various cocktails of compounds used to represent RIST. Could the authors explain in more detail how these combinations were determined in monolayer cells? I realise they have cited 2 papers but this is not enough, please explained this briefly here (any information about the combined toxicities of these compounds on the 5 cell lines?).

In figure 4, the combination of 2 drugs has unanimously reduced sphere viability. The schematic in figure 7 should be moved up here since it was not immediately clear to me that R+D represented pre-treatment which was then topped up with the I+T.

Please provide uncropped images for your WBs.

Figure 5 shows a reduction in the activated form (phosphorylated proteins such as p-p70S6k, P-4E-BP and SRC) compared to the non-phosphorylated forms.  Also in figure 6, the pre-treatment lowers CXCR4. In figure 5b, there is a difference between 2D and 3D culture with respect to CXCR4 levels. Please do these experiments for the other CSCs (NANOG, BMI1) and report the results.

In figure 7 the authors repeat the viability tests, for now, the full treatment and see a decrease in viability compared to pre-treatment. Since the plots are shown in two different figures, it is hard to judge how much greater the loss of viability has been compared to pre-treatment. Also, one issue with figures 4 and 7 and the cross-comparison between them is that the spheres generated are in disparate periods of time and perhaps the comparison may not be that simple since time is also a factor (3 and 7 days of incubation, respectively).

In figure 7, please repeat the experiments shown in Figures 5 and 6, now for the full treatment condition. 

The leap to La is not made clear. How did the authors arrive at La? It’s a gene linked to an unfavourable outcome in NB and is downregulated when using pre-treatment drugs. Please show what happens to La when the full treatment is used. Please show this experiment. 

Author Response

Please find my Point-by-Point Response to Comments of Reviewer 1 attached

Reviewer 2 Report

The authors established a spheroid cell culture model for neuroblastoma.
The study’s results demonstrate the efficiency of using the spheroid model for preclinical testing of new drugs. The increased expression of cancer stem cell markers and heightened T389 phosphorylation of La protein observed in the spheroids further solidify the authors' work.

The overall work is well presented and the importance of neuroblastoma tumor cultures is well understood.

However the authors should better explain how they obtain the spheroids.  In particular, it would be necessary to better motivate and explain the reasons that led to the choice of different 3D culture times for the different cell lines.
Different 3d culture times were used for the different NB lines. Was this done to achieve comparable spheroid sizes?

The authors could better characterize the spheroids for the number of cells that compose them and for the viability of the same. Even just by breaking them down and counting the number of cells and measuring their viability.

Improving these aspects would allow a more homogeneous and standardizable use of the 3D cultures of Neuroblastoma obtainable with the method described in the article.

Author Response

Please find my Point-by-Point Response to Comments of Reviewer 2 attached

Round 2

Reviewer 1 Report

The authors have addressed my comments.